# Chemotherapy and Survival in Patients with Primary High-Grade Extremity and Trunk Soft Tissue Sarcoma

**DOI:** 10.3390/cancers12092389

**Published:** 2020-08-24

**Authors:** Danielle S. Graham, Ritchell van Dams, Nicholas J. Jackson, Mykola Onyshchenko, Mark A. Eckardt, Benjamin J. DiPardo, Scott D. Nelson, Bartosz Chmielowski, Jacob E. Shabason, Arun S. Singh, Fritz C. Eilber, Anusha Kalbasi

**Affiliations:** 1Division of Surgical Oncology, Department of Surgery, University of California, Los Angeles, CA 90095, USA; DSGraham@mednet.ucla.edu (D.S.G.); MEckardt@mednet.ucla.edu (M.A.E.); BDipardo@mednet.ucla.edu (B.J.D.); FCEilber@mednet.ucla.edu (F.C.E.); 2Department of Radiation Oncology, University of California, Los Angeles, CA 90095, USA; RVanDams@mednet.ucla.edu; 3Department of Medicine Statistics Core, David Geffen School of Medicine, University of California, Los Angeles, CA 90095, USA; NJJackson@mednet.ucla.edu; 4Division of Hematology-Oncology, Department of Medicine, Harbor-UCLA Medical Center, The Lundquist Institute, Torrance, CA 90502, USA; MOnyshchenko@dhs.lacounty.gov; 5Department of Surgery, Yale School of Medicine, New Haven, CT 06519, USA; 6Department of Surgery, Greater Los Angeles Veterans Health Administration, Los Angeles, CA 90073, USA; 7Department of Pathology, University of California, Los Angeles, CA 90095, USA; SDNelson@mednet.ucla.edu; 8Jonsson Comprehensive Cancer Center, University of California, Los Angeles, CA 90095, USA; BChmielowski@mednet.ucla.edu (B.C.); ASingh@mednet.ucla.edu (A.S.S.); 9Division of Hematology-Oncology, Department of Internal Medicine, University of California, Los Angeles, CA 90095, USA; 10Abramson Family Cancer Research Institute, Perelman School of Medicine, University of Pennsylvania, Philadelphia, PA 19106, USA; JACOB.SHABASON@pennmedicine.upenn.edu; 11Department of Radiation Oncology, Perelman School of Medicine, University of Pennsylvania, Philadelphia, PA 19104, USA

**Keywords:** soft tissue sarcoma, chemotherapy, radiation, surgery, National Cancer Database

## Abstract

The use of upfront chemotherapy for primary localized soft tissue sarcoma (STS) of the extremity and trunk is debated. It remains unclear if chemotherapy adds clinical benefit, which patients are likely to benefit, and whether the timing of therapy affects outcomes. We used the National Cancer Database (NCDB) to examine the association between overall survival (OS) and chemotherapy in 5436 patients with the five most common subtypes of STS with primary disease localized to the extremity or trunk, mirroring the patient population of a modern phase 3 clinical trial of neoadjuvant chemotherapy. We then examined associations between timing of multi-agent chemotherapy (neoadjuvant or adjuvant) and OS. We used a Cox proportional hazards model and propensity score matching (PSM) to account for covariates including demographic, patient, clinical, treatment, and facility factors. In the overall cohort, we observed no association between multi-agent chemotherapy or its timing and improved OS. Multi-agent chemotherapy was associated with improved OS in several subgroups, including patients with larger tumors (>5 cm), those treated at high-volume centers, or those who received radiation. We also identified an OS benefit to multi-agent chemotherapy among the elderly (>70 years) and African American patients. Multi-agent chemotherapy was associated with improved survival for patients with tumors >5 cm, who receive radiation, or who receive care at high-volume centers. Neither younger age nor chemotherapy timing was associated with better outcomes. These ‘real-world’ findings align with recent randomized trial data supporting the use of multi-agent chemotherapy in high-risk patients with localized STS.

## 1. Introduction

Surgery, with or without radiation, remains the mainstay of treatment for adult patients with primarily localized extremity and trunk soft tissue sarcoma (STS). However, there is conflicting evidence regarding the impact of chemotherapy on survival for these patients [1,2,3,4,5,6,7,8,9,10,11]. Clinical practice guidelines reflect the equivocal nature of the data and present chemotherapy as a treatment option for high-grade disease [12,13].

Thus, a treating physician must not only decide whether to offer chemotherapy, but also whether to use single- or multi-agent chemotherapy and whether to use a neoadjuvant or adjuvant approach. This is reflected in the wide variation in practice patterns [14]. While there are data to support a survival benefit associated with the use of multi-agent rather than single-agent chemotherapy regimens [6,9,11], neoadjuvant and adjuvant chemotherapy have not been directly compared.

Due to the rarity of sarcoma, large randomized controlled trials are technically difficult, requiring significant time and resources for accrual. National cancer registries can be leveraged to address important clinical questions for rare diseases. The National Cancer Database (NCDB) is a joint project of the Commission on Cancer (CoC) of the American College of Surgeons and the American Cancer Society. The NCDB provides a unique opportunity to address these questions with a larger patient population [15].

We sought to determine the association between the use of chemotherapy—including multi-agent and neoadjuvant regimens—on survival in the treatment of adult patients with primary high-grade extremity and trunk STS using the NCDB. We also sought to define the subgroup of patients that may benefit most from the use of chemotherapy.

## 2. Results

### 2.1. Characteristics of Study Cohort

Our analysis included 5436 patients. Figure 1 shows the number of patients included or excluded according to criteria described in the Methods. Table 1 summarizes the characteristics of the study cohort. Chemotherapy was administered to 18% (*n* = 964) of the study population. Of the patients receiving chemotherapy with data available on number of chemotherapy agents (*n* = 916), multi-agent chemotherapy was administered to 90% (*n* = 827) and single-agent chemotherapy was administered to 10% (*n* = 89) of patients. Of patients receiving chemotherapy with data available on timing (*n* = 699), neoadjuvant chemotherapy was administered to 59% (*n* = 410) of patients undergoing chemotherapy. Factors associated with overall, multi-agent, and neoadjuvant chemotherapy use are summarized in Appendix A.

### 2.2. Association between Chemotherapy and Overall Survival

There was no association between chemotherapy use and overall survival (OS) in the primary patient cohort. Using a Cox proportional hazards model that adjusted for demographic, patient, clinical, treatment, and facility factors, there was no reliable difference in OS with the use of chemotherapy; hazard ratio (HR) 0.96; 95% confidence interval (CI), 0.85–1.09. Propensity score matching (PSM) demonstrated the same result (HR 0.94; 95% CI, 0.81–1.10) (Figure 2A).

### 2.3. Association between Single- and Multi-Agent Chemotherapy and Overall Survival

In the fully adjusted Cox proportional hazards model, there was no reliable difference in OS with the use of multi-agent chemotherapy when compared to no chemotherapy in the overall cohort (HR, 0.90; 95% CI, 0.78–1.03). PSM demonstrated the same result (Figure 2B).

Consistent with prior meta-analyses, a Cox proportional hazards model demonstrated decreased OS survival associated with single-agent chemotherapy use when compared to no chemotherapy (HR 1.49; 95% CI 1.06–2.10). PSM demonstrated similar results (Figure 2C).

### 2.4. Association between Neoadjuvant Chemotherapy and Overall Survival

There was no association between neoadjuvant chemotherapy use and OS in the overall cohort. Using a Cox proportional hazards model, there was no difference in OS with the use of neoadjuvant chemotherapy (HR, 1.13; 95% CI, 0.85–1.50). PSM demonstrated similar results (HR, 1.22; 95% CI, 0.92–1.63) (Figure 2D).

### 2.5. Subgroup Analyses

Due to a small sample size, the patients treated with single-agent chemotherapy were excluded from subgroup analyses. Subgroup analyses examining neoadjuvant versus adjuvant chemotherapy were not conducted due to the small sample size. The summary of interaction terms used to generate the subgroups is found in Appendix A, and a comprehensive summary of the results of subgroup analyses is found in Appendix A.

Statistically significant results of the multi-agent chemotherapy subgroup analyses are summarized in Figure 3. For patients with larger tumors (both >5 cm and >10 cm) the use of multi-agent chemotherapy was associated with improved survival. Multi-agent chemotherapy was also associated with improved survival in patients treated at high-volume facilities and patients treated with neoadjuvant or adjuvant radiation. Surgical margin status and histology did not significantly impact the association of multi-agent chemotherapy on survival. For deep tumors, there was a trend toward an association between multi-agent chemotherapy and improved survival, but this did not reach significance (HR, 0.83; 95% CI 0.67, 1.03; *p* < 0.10).

Based on the findings above, we defined a subgroup of patients who were most likely to benefit from multi-agent chemotherapy: those with larger tumors (>5 cm) treated with radiation at high-volume facilities. A Kaplan–Meier analysis of these patients indeed indicated improved OS associated with the use of multi-agent chemotherapy among the subgroups individually and as a combined cohort (Figure 4A–E).

Of note, younger patients were not more likely to benefit from multi-agent chemotherapy; in fact, we observed that for patients ≥70 years of age and for African American patients, multi-agent chemotherapy was associated with improved survival. However, based on small sample size, these factors were not considered in defining a subgroup of patients most likely to benefit from chemotherapy and the subsequent Kaplan–Meier analysis in Figure 4.

## 3. Discussion

We examined the association between chemotherapy—including number of agents and timing—with overall survival for adult patients with primary high-grade extremity and trunk STS. Our study had three main findings. First, the subgroup analysis identified a subset of patients with large (>5 cm) tumors treated with radiation at high-volume facilities for whom multi-agent chemotherapy was associated with improved survival. Second, there was no association between overall chemotherapy use, multi-agent chemotherapy use, or the timing of chemotherapy and OS in the overall study population. Lastly, there was no association between younger age and improved OS in our cohort; in fact, for the limited number of elderly (≥70 years) and African American patients in our cohort, treatment with multi-agent chemotherapy was associated with improved survival.

Randomized trials and meta-analyses have provided conflicting evidence regarding the benefit of chemotherapy in the treatment of adult patients with high-grade STS [1,2,3,4,5,6,7,8,9,10,11]. These different results may be in part because study inclusion criteria and chemotherapy regimens varied widely. In our study, the finding that there was no association between chemotherapy use and OS in an unselected patient cohort is thus unsurprising.

While our study found no association between multi-agent chemotherapy and OS in the overall study cohort, subgroup analysis highlighted its potential benefit for patients with larger tumors (>5 cm), those that also received radiation, and those treated at high-volume facilities. This is consistent with the design of previous RCTs that selected patients with large (>5 cm) tumors, based on the principle that these patients are most likely to benefit from chemotherapy [1,2,8,16,17,18], and is also consistent with the findings of an earlier NCDB analysis [19]. Further, treatment at high-volume facilities has been associated with improved survival in multiple cancers, including sarcoma [20,21,22,23].

While radiation therapy does not impact survival [24], our results suggest that the effect of chemotherapy on survival may be pronounced in patients with optimal local control. This is supported by an analysis of “high-risk” STS patients, defined by criteria including size ≥5 cm and high-grade, that identified an OS benefit with the addition of neoadjuvant chemotherapy to radiation [25]. A recent randomized trial compared standard anthracycline plus ifosfamide (AIM) neoadjuvant chemotherapy to histology-tailored (HT) neoadjuvant chemotherapy for localized extremity or trunk STS in the five histologic subtypes included in our analysis [26]. Use of HT neoadjuvant chemotherapy was associated with significantly worse OS compared to AIM neoadjuvant chemotherapy, suggesting a survival benefit to the use of neoadjuvant AIM chemotherapy. A subgroup analysis identified the largest survival benefit in patients with a nomogram-predicted OS of <60% [27], which is a high-risk group that typically includes patients of older age with larger, higher-grade tumors. Notably, both study arms included the use of radiation therapy for approximately 80% of their patients [26]. Taken as a whole, these findings support our identification of a survival benefit to neoadjuvant chemotherapy among patients with large tumors who receive radiation therapy.

To date, no randomized controlled trials have directly compared neoadjuvant to adjuvant chemotherapy in sarcoma. In our study chemotherapy timing was not associated with overall survival. Subgroup analyses, aimed to determine which patients may differentially benefit from neoadjuvant or adjuvant therapy, were largely underpowered. In practice, the decision on chemotherapy timing may be driven less by survival outcomes, but practical considerations such as the need for surgical downstaging.

One finding on our subgroup analysis that may be worth further exploration is the association with increased OS among elderly and African American patients receiving multi-agent chemotherapy. Caution is needed in interpreting this result, as the subgroup sizes are small and may be especially prone to selection biases. For example, elderly patients selected to receive multi-agent chemotherapy are likely to be a cohort selected for highest performance status or tolerance of chemo, which may be a confounding factor associated with survival. Nevertheless, this result highlights the importance of consideration of multi-agent chemotherapy for all patients regardless of chronologic age, and puts into question the practice of using younger chronological age as a selection tool for chemotherapy utilization [28]. With respect to our findings on race, prior research has identified decreased overall survival among African American patients with soft tissue sarcoma as compared to White or Asian patients [29]. Although they receive chemotherapy at similar rates as Whites, African American patients are more likely to have larger (>5 cm) or higher-grade tumors [29], which may explain the OS benefit of multi-agent chemotherapy seen for this subgroup in our analysis.

In a meta-analysis directly comparing multi-agent and single-agent regimens, multi-agent therapy was associated with improved OS [9]. In another meta-analysis, single-agent chemotherapy was associated with no difference in OS, while multi-agent chemotherapy was associated with improved OS, with both compared to no chemotherapy [6]. Our results largely align with the findings from those meta-analyses. Our analysis of single-agent chemotherapy, performed for internal validation of the data set, confirmed that these patients have worse outcomes, particularly those with smaller tumors with lower risk of metastatic dissemination. This finding does not immediately suggest that single-agent chemotherapy is without utility. The relative underperformance of single-agent chemotherapy may in part be due to selection bias, as patients treated with single-agent chemotherapy may have worse underlying performance status otherwise not captured by the NCDB. Use of single-agent chemotherapy may be beneficial in this cohort with poor performance status, and comparisons to cohorts who did not receive chemotherapy may reflect a patient population with more favorable prognostic factors, such as superficial, small tumors, who were a priori deemed to see little benefit from chemotherapy.

This study has limitations. First, our study is limited to CoC-accredited facilities in the United States. CoC facilities tend to be larger, are more frequently located in urban areas, have higher surgical volume, and more frequently treat patients with cancer when compared to non-CoC accredited facilities [30,31]. Second, while the NCDB is fairly comprehensive, it only represents approximately 70% of new cancer diagnoses annually [15]. Together, these factors may limit the generalizability of our findings. Third, while this study is retrospective and hypothesis-generating, causal links cannot be established. Fourth, there are no data regarding referral patterns or patient preferences in the NCDB. As such, there may be factors associated with care at high-volume centers that are not captured in this data. Specifically, treatment at a high-volume center may be associated with several confounders that may improve patient prognosis, including improved access to clinical trials, more comprehensive health care coverage, higher socioeconomic status, and more. Additionally, there may be variability in how various centers code clinical data, particularly what is considered chemotherapy versus systemic therapy. Fifth, although efforts have been made to control for potential covariates through Cox modeling and propensity score matching, limitations of the dataset mean certain covariates, such as performance status, are unavailable for adjustment. Finally, the NCDB does not record data on disease-free or metastasis-free survival, and thus we were limited to overall survival as a surrogate for the efficacy of chemotherapy.

## 4. Materials and Methods

### 4.1. Data Source and Cohort Definition

In this analysis, we used data from the NCDB from 2004 to 2016. The NCDB is sponsored by the American College of Surgeons and the American Cancer Society. It includes patient data from over 1500 CoC-accredited facilities in the United States and represents over 70% of new cancer cases in the United States [15].

From the initial patients included in the soft tissue sarcoma NCDB dataset, we selected patients with high-grade tumors, including the following histologies: myxoid liposarcoma, synovial sarcoma, malignant peripheral nerve sheath tumor (MPSNT), leiomyosarcoma, and undifferentiated pleomorphic sarcoma (UPS). We chose these histologies as they represent the five most common subtypes of sarcoma and account for 80% of all cases [16]. This cohort also reflects a similar population of patients (high-grade, large primary STS with these five subtypes) in a recent international randomized trial comparing histology-tailored versus conventional regimen neoadjuvant chemotherapy [26]. We further excluded pediatric patients (<18 years), low or unknown grade tumors, and patients with positive nodes or distant metastases.

From this population, we excluded patients with primary site of disease other than trunk or extremity, those who underwent palliative treatment, those who underwent radiation therapy outside the range of 40–75 Gy or with an unknown dose, and those who experienced delays in care above the 85^th^ percentile (defined as the time between diagnosis and any treatment modality or between different modalities, including surgery, radiation, and chemotherapy) (Appendix A). Finally, we excluded patients who did not undergo surgery or whose surgery status was unknown, patients whose physicians recommended chemotherapy but did not undergo chemotherapy because of extensive comorbidities or premature death, patients who did not have any histologic confirmation of malignancy, patients without survival data, and those who did not have chemotherapy data available. This resulted in a final cohort of 5436 patients.

For the analysis of the impact of chemotherapy use on survival, all 5436 patients were included. Of the 964 patients who underwent chemotherapy, data on the number of agents used were available for 916. Thus, 5388 patients were included in the analysis of the association between number of chemotherapy agents and survival. Finally, data regarding the timing of chemotherapy were collected starting in 2006 and were available for 699 patients who underwent multi-agent chemotherapy treatment. As a result that our data suggested that single-agent chemotherapy was associated with decreased OS, we excluded these patients from our analysis of the association between neoadjuvant versus adjuvant chemotherapy use and survival (Figure 1).

The data used in the study are derived from a de-identified NCDB dataset. The American College of Surgeons and the Commission on Cancer have not verified and are not responsible for the analytic or statistical methodology employed, or the conclusions drawn from these data by the investigator.

### 4.2. Analytic Variables

The primary aim was to determine the association between OS and the use of multi-agent chemotherapy in adult patients with primary high-grade extremity and trunk STS. A separate analysis of the association between OS and single-agent chemotherapy was performed to internally validate the data set as prior research has identified that single-agent regimens are not associated with the same improvements in OS as multi-agent regimens [6,9]. Secondary aims included determining OS associated with adjuvant versus neoadjuvant chemotherapy, as well as determining the subgroup of patients that may benefit from chemotherapy. Covariates included demographic, patient, clinical, treatment, and facility factors, specifically age; sex; race/ethnicity; insurance status; income; county size; facility location, type, and volume; distance to treatment, Charlson–Deyo Comorbidity Score; histology; primary site, size, and depth; surgical margins; and use of radiation therapy.

Annual sarcoma surgical volume was analyzed in the initial NCDB population (*n* = 106,822) before any of the exclusions listed above to determine high-volume versus low-volume facilities [14]. The cut-off was the 99th percentile annual surgical case volume (Appendix A), which has been used in prior studies [32,33]. Distance from patient home to treatment facility was analyzed in the final study population. Distances were grouped based on relatively equal group size and presumed practical relevance (Appendix A).

### 4.3. Statistical Analysis

Univariate descriptive statistics (means, standard deviations, and relative frequencies) were used to summarize baseline study cohort characteristics. Unadjusted associations of chemotherapy with OS were graphically assessed using Kaplan–Meier survival curves. Differences in demographic and clinical factors between chemotherapy groups were assessed using independent samples t-tests and Pearson chi-square tests, as appropriate. Variables that were statistically significant (*p* < 0.05) in these bivariate analyses or that were standard covariates in the literature, such as gender and age, were included as covariates in adjusted models. A Cox proportional hazards model was used to determine the association between chemotherapy use and OS, adjusting for the aforementioned covariates. Reference groups were the groups with the largest number of patients in each category. Huber-White standard errors were used to account for clustering by facility. We additionally performed PSM using a 1:1 nearest neighbor matching algorithm to compare against the results of the Cox proportional hazards model. Results are shown as hazard ratios (HR) with 95% confidence intervals (CI). The above analysis was repeated on each subgroup to determine survival associated with number of chemotherapy agents as well as neoadjuvant chemotherapy.

We performed subgroup analyses to determine the cohort of patients that may differentially benefit from the use of chemotherapy. Subgroups were selected using both data-driven and evidence-driven approaches. Specifically, differential associations between chemotherapy use and OS for subgroups were assessed based on a chemotherapy-by-subgroup interaction term in the Cox model. Additional subgroups were based on tumor size, tumor depth, and facility volume, as prior literature has demonstrated an association between these factors and survival [1,2,8,16,17,20,21,22,23]. Stratified analyses were used to examine the simple effects of chemotherapy on survival for subgroups with a significant interaction term. This analysis was repeated to determine survival associated with number of chemotherapy agents and neoadjuvant chemotherapy in subgroups of patients. The subgroups of patients with statistically significant results were selected, and a Kaplan–Meier analysis was performed to determine the association between multi-agent chemotherapy and survival in each subset of patients, as well as an aggregate of subsets.

Statistical analysis was performed using Stata (StataCorp. 2017. *Stata Statistical Software: Release 15*. College Station, TX, USA: StataCorp LLC).

## 5. Conclusions

In summary, multi-agent chemotherapy was associated with improved survival in selected subsets of patients with high-risk disease, including those with large tumors, those treated with radiation, and those treated at high-volume centers. These real world findings are consistent with data from recent randomized trials of neoadjuvant chemotherapy in this setting. We did not find an association between chemotherapy timing on survival, nor was there a selective advantage to chemotherapy in younger patients. In the absence of more instructive guidelines for primary extremity and trunk STS, these data highlight how physicians’ decisions on patient selection drive the impact of chemotherapy.

## Figures and Tables

**Figure 1 cancers-12-02389-f001:**
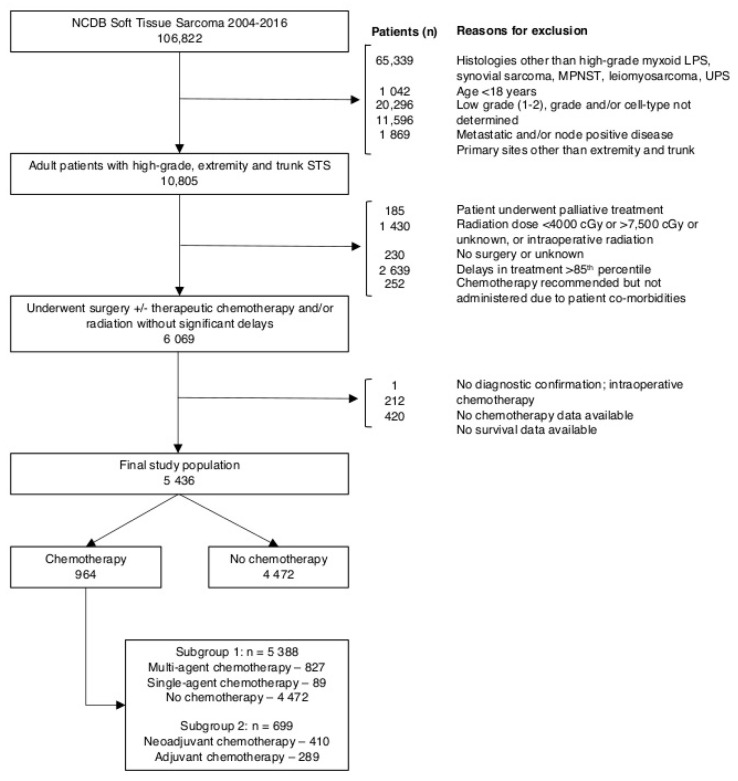
Consolidated Standards of Reporting Trials (CONSORT) diagram showing definition of study cohort from NCDB STS database. NCDB = National Cancer Database; STS = soft tissue sarcoma.

**Figure 2 cancers-12-02389-f002:**
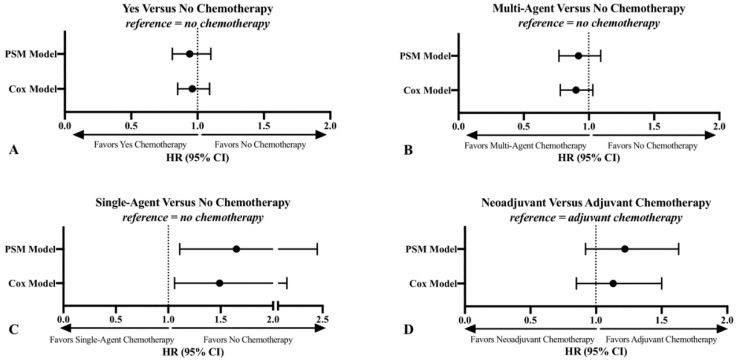
Cox proportional hazards model and PSM results of the association between chemotherapy use (**A**), multi-agent chemotherapy use (**B**), single-agent chemotherapy use (**C**), and neoadjuvant chemotherapy use (**D**) with OS in patients with primary high-grade extremity and trunk STS in the NCDB from 2004–2016. CI = confidence interval; HR = hazard ratio; NCDB = National Cancer Database; OS = overall survival; PSM = propensity score matching; STS = soft tissue sarcoma.

**Figure 3 cancers-12-02389-f003:**
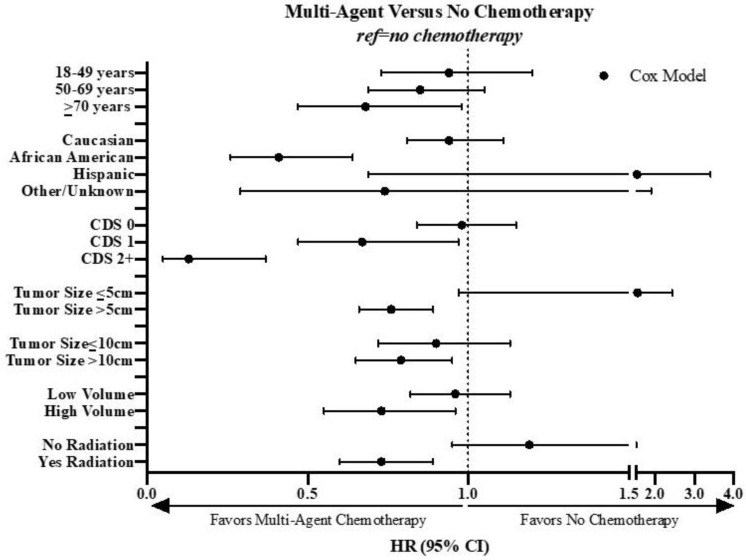
Cox proportional hazards model and PSM results of the subgroup analysis of the association between multi-agent chemotherapy and OS in patients with primary high-grade extremity and trunk STS in the NCDB from 2004 to 2016. CDS = Charlson–Deyo Score; CI = confidence interval; HR = hazard ratio; NCDB = National Cancer Database; OS = overall survival; PSM = propensity score matching; STS = soft tissue sarcoma.

**Figure 4 cancers-12-02389-f004:**
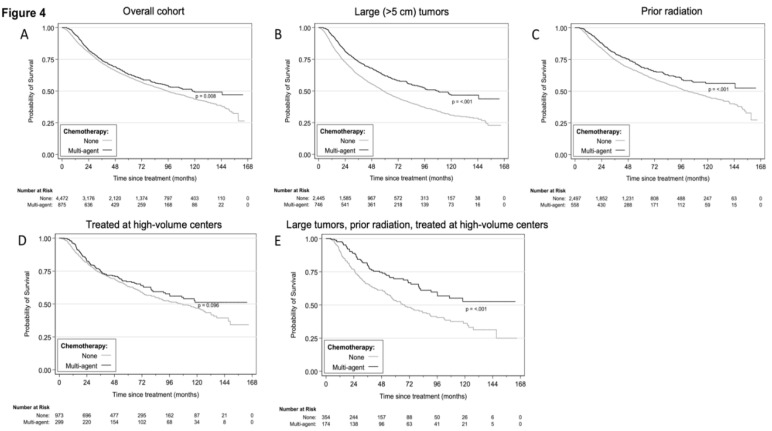
Kaplan–Meier analysis demonstrating the association between multi-agent chemotherapy use and OS in the overall cohort (**A**), in patients with large (>5 cm) tumors (**B**), with prior radiation (**C**), treated at high-volume centers (**D**), and in patients with primary high-grade extremity and trunk STS who had large (>5 cm) tumors and were treated with radiation at high-volume centers (**E**). Study sample derived from the NCDB from 2004 to 2016. NCDB = National Cancer Database; OS = overall survival; STS = soft tissue sarcoma.

**Table 1 cancers-12-02389-t001:** Characteristics of study cohort of adult patients with primary high-grade extremity and trunk STS in the NCDB from 2004 to 2016. NCDB = National Cancer Database; STS = soft tissue sarcoma.

Characteristic	No.	Percent
**Total**	5436	
**Sex**		
Male	2929	53.9
Female	2507	46.1
**Age (years)**		
18–49	1332	24.5
50–69	2128	39.1
>70	1976	36.4
**Race/Ethnicity**		
Caucasian	4390	80.8
African American	507	9.3
Hispanic	283	5.2
Other and Unknown	256	4.7
**Insurance status**		
Commercial	2538	46.7
Medicare	2298	42.3
Medicaid	258	4.7
Uninsured	180	3.3
Other Government & Unknown	162	3.0
**Income**		
<$40,277	897	16.5
$40,227–50,353	1220	22.4
$50,354–63,332	1270	23.4
>$63,333	1949	35.9
Unknown	100	1.8
**County size**		
Metropolitan	4377	80.5
Urban	824	15.2
Rural	98	1.8
Unknown	137	2.5
**Facility location**		
East	963	17.7
South	1762	32.4
Central	1301	23.9
West	732	13.5
Unknown	678	12.5
**Facility type**		
Non-Academic	2143	39.4
Academic	2615	48.1
Unknown	678	12.5
**Facility volume**		
Low volume	4150	76.3
High volume	1286	23.7
**Distance to treatment**		
0–10 miles	2036	37.5
10–30 miles	1563	28.8
30–100 miles	1188	21.9
100+ miles	619	11.4
Unknown	30	0.6
**Charlson–Deyo Comorbidity Score**		
0	4438	81.6
1	802	14.8
2+	196	3.6
**Histology**		
High-grade myxoid LPS	313	5.8
Synovial sarcoma	473	8.7
MPNST	296	5.4
Leiomyosarcoma	1090	20.1
UPS	3264	60.0
**Primary Site**		
Extremity	5027	92.5
Trunk	409	7.5
**Size**		
≤5 cm	1881	34.6
5.1–10 cm	1816	33.4
10.1–15 cm	837	15.4
>15 cm	608	11.2
Unknown	294	5.4
**Depth**		
Superficial	1410	25.9
Deep	2220	40.8
Unknown	1806	33.2
**Surgical Margins**		
Negative	4401	81.0
Positive	814	15.0
Unknown	221	4.1
**Radiation Therapy**		
No	2323	42.7
Yes	3113	57.3

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
