# Peer review of "Chemotherapy and Survival in Patients with Primary High-Grade Extremity and Trunk Soft Tissue Sarcoma"

_cancers, 2020, doi:10.3390/cancers12092389_

Round 1

Reviewer 1 Report

There is ongoing controversy as to the role of chemotherapy in localized sarcomas. As noted by the authors, there are many studies with mixed results- likely confounded by mixed histologies, alternate chemo dosing, and other clinical factors.  Consensus is growing that a subgroup may have marginal benefit from chemo, including the highest risk patients and selected chemo-sensitive histologies.

In this manuscript the authors identified over 5000 sarcoma pts w localized trunk/extremity disease in the NCDB, of which 18% received chemo.

In this analysis, the author’s findings mirror evidence in some of the larger clinical trials, that larger tumors drive the benefit.  There were also associations with high-volume centers (not surprising given the specialized nature of this disease), age, race, and radiation (may be influenced by size i.e. larger tumors will be radiated, standard of care). 

Overall this paper is well-written and appropriate for publication.

Most of my comments are minor

  • Page 2 line 61 ‘Due to the rarity of sarcoma, large RCT are not typically feasible’ – I think some readers would have issue with that statement given that there are several large RCT’s that have recently completed and/or are recruiting.
  • Did you analyze by histology? I understand the numbers are limited but it is an important point as there is a difference in chemo sensitivity in say synovial v MPNST. If the numbers are too small, you should at least comment on this issue.
  • Similarly, is there a breakdown of trunk v. extremity? Superficial v. deep?
  • Are you able to determine if most of the radiated tumors were indeed large tumors? It may explain the association there
  • Figure 4 needs titles above each curve. Most readers will skim and not read the legend. If above each curve you label ‘sarcomas >5 cm, prior radiation, high-vol center etc. it would present more clearly.

Author Response

Dear Reviewer 1,

Thank you for your feedback, both positive and constructive. We sincerely appreciate your time. Each of your comments has been addressed in the letter below (see bold type following each comment) as well as in our manuscript, where applicable.

Chemotherapy and Survival in Patients with Primary High-Grade Extremity and Trunk Soft Tissue Sarcoma

Cancers-867102

Reviewer 1

There is ongoing controversy as to the role of chemotherapy in localized sarcomas. As noted by the authors, there are many studies with mixed results- likely confounded by mixed histologies, alternate chemo dosing, and other clinical factors.  Consensus is growing that a subgroup may have marginal benefit from chemo, including the highest risk patients and selected chemo-sensitive histologies.

In this manuscript the authors identified over 5000 sarcoma pts w localized trunk/extremity disease in the NCDB, of which 18% received chemo.

In this analysis, the author’s findings mirror evidence in some of the larger clinical trials, that larger tumors drive the benefit.  There were also associations with high-volume centers (not surprising given the specialized nature of this disease), age, race, and radiation (may be influenced by size i.e. larger tumors will be radiated, standard of care).

Overall this paper is well-written and appropriate for publication.

Most of my comments are minor

Page 2 line 61 ‘Due to the rarity of sarcoma, large RCT are not typically feasible’ – I think some readers would have issue with that statement given that there are several large RCT’s that have recently completed and/or are recruiting.

We have edited the language to the following, “Due to the rarity of sarcoma, large randomized controlled trials are technically difficult, requiring significant time and resources for accrual.”

Did you analyze by histology? I understand the numbers are limited but it is an important point as there is a difference in chemo sensitivity in say synovial v MPNST. If the numbers are too small, you should at least comment on this issue.

Page 6, line 125 now includes our finding that histology had no significant impact on the association of multi-agent chemotherapy and survival.

Similarly, is there a breakdown of trunk v. extremity? Superficial v. deep?

Are you able to determine if most of the radiated tumors were indeed large tumors? It may explain the association there

Page 6, lines 126-128 mention a trend to significance between deep tumors and a benefit to multi-agent chemotherapy. This was included in the analysis as there have been reports that this variable is associated with outcomes. Extremity v. trunk was not analyzed as a separate variable as the large majority of patients had tumors in the extremity and this variable has not been reproducibly associated with outcome in the literature. While the factors associated with radiation therapy were not separately analyzed (this is analyzed in a separate manuscript that was recently published in JNCCN [1]), the effect of radiation was observed after adjusting for other variables, including tumor size, on multivariate analysis.

Figure 4 needs titles above each curve. Most readers will skim and not read the legend. If above each curve you label ‘sarcomas >5 cm, prior radiation, high-vol center etc. it would present more clearly.

Figure 4 has been amended to include caption titles above each subfigure.

References 

  1. Graham, D.S.; Onyshchenko, M.; Eckardt, M.A.; DiPardo, B.J.; Venigalla, S.; Nelson, S.D.; Chmielowski, B.; Singh, A.S.; Shabason, J.E.; Eilber, F.C.; et al. Low Rates of Chemotherapy Use for Primary, High-Grade Soft Tissue Sarcoma: A National Cancer Database Analysis. J. Natl. Compr. Cancer Netw. 2020, 18, 1055–1065, doi:10.6004/jnccn.2020.7553.

Reviewer 2 Report

The authors have presented a study on analysis of  the National Cancer Database (NCDB) to find  an association between overall survival and chemotherapy the patients with Soft Tissue Sarcoma (STS).

The study was designed in a good way and methods and results are presented in a clear way.

I believe the results are interesting and useful for the field of  STS. 

My only concern is about the small fonts on the figures which are hard to read. 

Author Response

Dear Reviewer 2,

Thank you for your feedback, both positive and constructive. We sincerely appreciate your time. Each of your comments has been addressed in the letter below (see bold type following each comment) as well as in our manuscript, where applicable.

Reviewer 2

The authors have presented a study on analysis of  the National Cancer Database (NCDB) to find  an association between overall survival and chemotherapy the patients with Soft Tissue Sarcoma (STS).

The study was designed in a good way and methods and results are presented in a clear way.

I believe the results are interesting and useful for the field of  STS.

My only concern is about the small fonts on the figures which are hard to read.

Thank you for raising this concern. The figures will be supplied in full resolution should be legible in both print and digital formats on final publication.

Reviewer 3 Report

Grahma and her co-authors have analysed the overall survival data in a large cohort of 5436 patients with the five most common subtypes of soft tissue sarcomas of the extremity and trunk, derived from NCDB database. For the cohort as a whole, no OS benefit associated with multi-chemotherapy could be found. However, using Cox proportional hazards mode and propensity score matching method, they have found some OS benefit associated with chemotherapy in several subgroups (larger tumour, treatment in high-volume centre, radiotherapy, elderly or African American patients).

Their findings in this “real-world” large patient’s cohort confirmed the results of several phase III clinical trials and underlined the importance of treatment in high-volume center, considering the complexity of multimodality treatment and adequate management of therapy-induced complications, and an individualized cancer care, especially in the quite inhomogeneous STS patients. Although it is a retrospective study, the large number of included patients may somehow compensate this weakness.

Several points should be addressed before considering this manuscript for publication.

  1. i) The authors have compared the subgroups “single- vs multi-agent chemotherapy”. It is important to know if the chemotherapy was applied sequentially or simultaneously to radiotherapy, since single-agent chemotherapy is normally used simultaneously to radiotherapy, rather than sequential application.
  2. ii) Two important phase III trials are missing in the manuscript, which should be added and discussed in the discussion part.

https://www.thelancet.com/article/S1470-2045(10)70071-1/fulltext

https://jamanetwork.com/journals/jamaoncology/fullarticle/2672386

Minor points:

Figure 1 is too small for a clear presentation.

Please explain the abbreviation “CDS” in the legend of figure 3.

Author Response

Dear Reviewer 3,

Thank you for your feedback, both positive and constructive. We sincerely appreciate your time. Each of your comments has been addressed in the letter below (see bold type following each comment) as well as in our manuscript, where applicable.

Reviewer 3

Grahma and her co-authors have analysed the overall survival data in a large cohort of 5436 patients with the five most common subtypes of soft tissue sarcomas of the extremity and trunk, derived from NCDB database. For the cohort as a whole, no OS benefit associated with multi-chemotherapy could be found. However, using Cox proportional hazards mode and propensity score matching method, they have found some OS benefit associated with chemotherapy in several subgroups (larger tumour, treatment in high-volume centre, radiotherapy, elderly or African American patients).

Their findings in this “real-world” large patient’s cohort confirmed the results of several phase III clinical trials and underlined the importance of treatment in high-volume center, considering the complexity of multimodality treatment and adequate management of therapy-induced complications, and an individualized cancer care, especially in the quite inhomogeneous STS patients. Although it is a retrospective study, the large number of included patients may somehow compensate this weakness.

Several points should be addressed before considering this manuscript for publication.

  1. i) The authors have compared the subgroups “single- vs multi-agent chemotherapy”. It is important to know if the chemotherapy was applied sequentially or simultaneously to radiotherapy, since single-agent chemotherapy is normally used simultaneously to radiotherapy, rather than sequential application.

This is an excellent point. The sequence of individual therapies is not explicitly defined in the National Cancer Database. However, the timeline of treatment initiation can be roughly ascertained for each treatment type based on a variable that defines the time from diagnosis to the time of treatment initiation. Furthermore, the timing of treatment completion is not indicated, so it is not possible to determine whether chemotherapy was completed prior to initiation of RT. Given the imprecise nature of this type of analysis, we think it is too difficult to draw clear conclusions about sequential versus concurrent therapy. In addition, the overall number of patients receiving single-agent chemotherapy is far lower, and may be associated with poor performance status and older age, that will confound results. Thus, while this is a great point that deserves investigation, this dataset is not best suited to answer this question.

Nonetheless, in a separate analysis of 2,359 soft tissue sarcoma patients treated with neoadjuvant chemotherapy and radiation, we determined the time between chemotherapy and RT using the diagnosis to treatment initiation time variable. The time between initiation of chemo and initiation of RT was a median of 29 days (range -85 to 188). This indicates that most patients begin with chemotherapy first, and receive at least ~1-2 cycles of chemotherapy prior to initiation of radiation.

  1. ii) Two important phase III trials are missing in the manuscript, which should be added and discussed in the discussion part.

https://www.thelancet.com/article/S1470-2045(10)70071-1/fulltext

https://jamanetwork.com/journals/jamaoncology/fullarticle/2672386

We thank the reviewers for highlighting these important manuscripts which describe the impact of hyperthermia on the effect of neoadjuvant chemotherapy. However, we are not certain that these manuscripts have any direct relatedness to the question of the effectiveness of chemotherapy – as all patients in these trials received chemotherapy. If the reviewer would explain why these manuscripts add to the discussion of the effectiveness of neoadjuvant chemotherapy we would be happy to reconsider.

Minor points:

Figure 1 is too small for a clear presentation.

Thank you for raising this concern. The figures will be supplied in full resolution should be legible in both print and digital formats on final publication.

Please explain the abbreviation “CDS” in the legend of figure 3.

Thank you for catching this error. We have included the explanation for the Charlson-Deyo Score in the figure legend on page 7, line 132.

Reviewer 4 Report

In this analysis, Graham et al do a thorough job analyzing the effect on outcomes with neo/adjuvant chemotherapy in high grade, trunk/extremity, soft tissue sarcoma from the NCDB database. They found that in the overall population there was no evidence of benefit for multi-agent chemotherapy but in the subgroup of patients with large tumors (>5cm) there was a significant benefit. Also intriguingly, the significant effect was driven by elderly patients, especially those > age 70. It is a well analyzed and well written study. Comments below.

  • In the Associations between single- and multi-agent chemo and oversall survival, it states that “As expected, a Cox proportional hazards model demonstrated decreased OS survival associated with single-agent chemotherapy use when compared to no chemotherapy”. Why is this as expected?
  • It is extraordinary that benefit of multi-agent chemo increases with age and is most pronounced in the over 70 years of age group as this is the group with the highest likelihood of developing complications for intensive therapy. What is the number of patients treated with chemo (and not) in each age group?
  • In the sub-group analysis there is evidence of benefit in tumor size >=5cm. However, is this a full multi-variate Cox model that takes into account all variables (i.e. Age, race, volume, radiation)? I would explore an additional analysis restricting to just patients with Tumor Size >=5 and running a full Cox model in order to get hazard ratios for each variable.
  • Figure 3 has a row for CDS 0, 1, 2+ but there is no definition of what CDS is. Is this performance status?
  • Figure 4A shows a significant p-value for the Kaplan meier =0.008 but in the text it is described as not significant OS difference. Please explain.
  • There needs to be stronger wording in the limitations that this is a retrospective study and thus there are inherent biases in the distribution of treatment. Most importantly, patients with good performance status were more likely to receive chemo and performance status is a very important predictor of prognosis. I do not see performance status used as a variable here and I assume it was likely not available.
  • One way to “control” for this inherent bias is to see if there are differences in institutional practices. If one institution prescribes adjuvant chemo and one does not then those institutions can be compared to see if there are differences in outcomes. Propensity score matching can be used to try to alleviate some of the differences in patient population between the two centers. Is the NCDB set up to do this?
  • There is no mention of subgroup analysis within each sarcoma histology. This is very important as sarcomas are different from each other. For instance, myxoid liposarcoma and synovial sarcoma are the two histologies with the highest sensitivity to chemo while MPNST is less so.
  • What were the chemotherapies administered? Was it the standard AIM regimen for most of the patients?
  • I would place Methods after Conclusions.

Author Response

Dear Reviewer 4,

Thank you for your feedback, both positive and constructive. We sincerely appreciate your time. Each of your comments has been addressed in the letter below (see bold type following each comment) as well as in our manuscript, where applicable.

Reviewer 4

In this analysis, Graham et al do a thorough job analyzing the effect on outcomes with neo/adjuvant chemotherapy in high grade, trunk/extremity, soft tissue sarcoma from the NCDB database. They found that in the overall population there was no evidence of benefit for multi-agent chemotherapy but in the subgroup of patients with large tumors (>5cm) there was a significant benefit. Also intriguingly, the significant effect was driven by elderly patients, especially those > age 70. It is a well analyzed and well written study. Comments below.

In the Associations between single- and multi-agent chemo and oversall survival, it states that “As expected, a Cox proportional hazards model demonstrated decreased OS survival associated with single-agent chemotherapy use when compared to no chemotherapy”. Why is this as expected?

Prior meta-analyses (described and referenced on page 9, lines 214-217) have found that multi-agent chemotherapy has been found to be associated a survival advantage as compared to single-agent chemotherapy. However, we have edited the text to remove this language and replace with “Consistent with prior meta-analyses…”

It is extraordinary that benefit of multi-agent chemo increases with age and is most pronounced in the over 70 years of age group as this is the group with the highest likelihood of developing complications for intensive therapy. What is the number of patients treated with chemo (and not) in each age group?

Supplementary Table 1 provides the breakdown of patients in each age category by chemotherapy regimen:

Age

Yes Chemotherapy n (%)

No Chemotherapy n (%)

[p-value] <0.001

Multi-agent chemotherapy n (%)

Single-agent chemotherapy n (%)

No chemotherapy n (%)

18-49yo

430 (32.3)

902 (67.7)

391 (29.7)

24 (1.8)

902 (68.5)

50-69yo

436 (20.5)

1,692 (79.5)

372 (17.7)

42 (2.0)

1,692 (80.3)

>70yo

98 (5.0)

1,878 (95.0)

64 (3.3)

23 (1.2)

1,878 (95.6)

In the sub-group analysis there is evidence of benefit in tumor size >=5cm. However, is this a full multi-variate Cox model that takes into account all variables (i.e. Age, race, volume, radiation)? I would explore an additional analysis restricting to just patients with Tumor Size >=5 and running a full Cox model in order to get hazard ratios for each variable.

We thank the reviewers for this comment. We have in fact included these results in the manuscript as part of the supplementary material (Supplementary Table 3B), where there is a substantial benefit to multi-agent chemotherapy (compared to no chemotherapy) on multi-variable analysis (in both the Cox model and propensity score matched models) for patients with tumors >5cm. For your convenience we have copied these results below: 

Multi-agent versus no chemotherapy
reference = no chemotherapy

Multi-agent chemotherapy
n (%)

No Chemotherapy
 n (%)

Cox model
(HR, 95% CI)

PSM
(HR, 95% CI)

<5cm

94 (5.0)

1,771 (94.2)

†1.54 (0.97, 2.43)

0.96 (0.54, 1.71)

>5cm

703 (21.8)

2,445 (76.0)

***0.76 (0.66, 0.89)

*0.84 (0.70, 0.99)

 Figure 3 has a row for CDS 0, 1, 2+ but there is no definition of what CDS is. Is this performance status?

Thank you for catching this error. We have included the explanation for the Charlson-Deyo Score in the figure legend on page 7, line 132.

Figure 4A shows a significant p-value for the Kaplan meier =0.008 but in the text it is described as not significant OS difference. Please explain.

The analyses in Figure 4 overall do not utilize the full Cox model. As a result, this K-M finding is representative of a univariate analysis of multi-agent chemotherapy on the overall cohort and has not been adjusted for all other covariates.

There needs to be stronger wording in the limitations that this is a retrospective study and thus there are inherent biases in the distribution of treatment. Most importantly, patients with good performance status were more likely to receive chemo and performance status is a very important predictor of prognosis. I do not see performance status used as a variable here and I assume it was likely not available.

Page 9, lines 241-243 now include the following language on limitations of our study, “Fifth, although efforts have been made to control for potential covariates through Cox modelling and propensity score matching, limitations of the dataset mean certain unmeasured covariates are unavailable for adjustment.”

One way to “control” for this inherent bias is to see if there are differences in institutional practices. If one institution prescribes adjuvant chemo and one does not then those institutions can be compared to see if there are differences in outcomes. Propensity score matching can be used to try to alleviate some of the differences in patient population between the two centers. Is the NCDB set up to do this?

The Cox model and PSM were used to evaluate treatment centers by volume, but individual-patient- and treatment-center-level data create sample sizes that are too small for generalizable statistical findings regarding differences in outcomes between these centers. However, in previous work we have used generalized estimating equations to account for clustering of outcomes within facility assuming an independent correlation structure [1]. We have conducted similar analysis for the sarcoma dataset (data not shown) with no impact on the data presented here.

 There is no mention of subgroup analysis within each sarcoma histology. This is very important as sarcomas are different from each other. For instance, myxoid liposarcoma and synovial sarcoma are the two histologies with the highest sensitivity to chemo while MPNST is less so.

Page 6, line 125 now includes our finding that histology had no significant impact on the association of multi-agent chemotherapy and survival.

What were the chemotherapies administered? Was it the standard AIM regimen for most of the patients?

Specific chemotherapy agents used for patients are not available in the NCBD dataset.

I would place Methods after Conclusions.

This manuscript format order was described by the Cancers author instructions and submission template.

References

  1. Kalbasi, A.; Swisher-McClure, S.; Mitra, N.; Sunderland, R.; Smaldone, M.C.; Uzzo, R.G.; Bekelman, J.E. Low rates of adjuvant radiation in patients with nonmetastatic prostate cancer with high-risk pathologic features. Cancer 2014, 120, 3089–3096, doi:10.1002/cncr.28856.

Reviewer 5 Report

This is a well written manuscript addressing the impact of chemotherapy on overall survival for patients with high grade extremity and trunk soft tissue sarcomas using retrospective analysis of NCBD data.

However, both their main findings, that chemotherapy improves survival in patients receiving radiation therapy (Chowdhary et al, PMID: 31490546) and patients with tumors larger than 5cm, ergo patients with high risk soft tissue sarcomas, (Movva et al., PMID: 26553766) have been shown in other studies using the same database. The latter study is not cited in this manuscript. Therefore, the novelty of their findings remain debatable.

Some minor points regarding the manuscript itself need to be addressed. Firstly, information on agents used for chemotherapy should be reported as they are available for 916 patients according to the authors.

Secondly, judging from the numbers at risk in Figure 4A, a survival benefit of patients receiving multiagent chemotherapy versus no chemotherapy is shown for the whole collective while the authors state that a benefit was only evident for several subgroups of patients that are presented in Figure 4B-E. This is confusing and needs clarification/correction.

Author Response

Dear Reviewer 5, 

Thank you for your feedback, both positive and constructive. We sincerely appreciate your time. Each of your comments has been addressed in the letter below (see bold type following each comment) as well as in our manuscript, where applicable.

Reviewer 5

This is a well written manuscript addressing the impact of chemotherapy on overall survival for patients with high grade extremity and trunk soft tissue sarcomas using retrospective analysis of NCBD data.

However, both their main findings, that chemotherapy improves survival in patients receiving radiation therapy (Chowdhary et al, PMID: 31490546) and patients with tumors larger than 5cm, ergo patients with high risk soft tissue sarcomas, (Movva et al., PMID: 26553766) have been shown in other studies using the same database. The latter study is not cited in this manuscript. Therefore, the novelty of their findings remain debatable.

Some minor points regarding the manuscript itself need to be addressed. Firstly, information on agents used for chemotherapy should be reported as they are available for 916 patients according to the authors.

Specific chemotherapy agents used for patients are not available in the NCBD dataset.

Secondly, judging from the numbers at risk in Figure 4A, a survival benefit of patients receiving multiagent chemotherapy versus no chemotherapy is shown for the whole collective while the authors state that a benefit was only evident for several subgroups of patients that are presented in Figure 4B-E. This is confusing and needs clarification/correction.

The analyses in Figure 4 overall do not utilize the full Cox model. As a result, this K-M finding is representative of a univariate analysis of multi-agent chemotherapy on the overall cohort and has not been adjusted for all other covariates.

Reviewer 6 Report

I enjoyed reading this well written and interesting manuscript.

  1. My major point relates to the challenges of pathological diagnosis. The authors should discuss in more detail the potential variations in pathological diagnosis without central pathology review. Could this have an impact on findings, particularly for pleomorphic sarcoma, leiomyosarcoma and MPNST? Was translocation status confirmed for all synovial and myxoid liposarcoma cases?
  2. The authors should define in more detail ‘high’ and ‘low’ volume centres.

Author Response

Dear Reviewer 6, 

Thank you for your feedback, both positive and constructive. We sincerely appreciate your time. Each of your comments has been addressed in the letter below (see bold type following each comment) as well as in our manuscript, where applicable.

Reviewer 6

I enjoyed reading this well written and interesting manuscript.

My major point relates to the challenges of pathological diagnosis. The authors should discuss in more detail the potential variations in pathological diagnosis without central pathology review. Could this have an impact on findings, particularly for pleomorphic sarcoma, leiomyosarcoma and MPNST? Was translocation status confirmed for all synovial and myxoid liposarcoma cases?

Data regarding specific pathologic diagnosis techniques are not available in the NCDB dataset. Regardless, Page 6, line 125 now includes our finding that histology had no significant impact on the association of multi-agent chemotherapy and survival.

The authors should define in more detail ‘high’ and ‘low’ volume centres.

Sarcoma-specific annual case volume is available on a per-institution level and a histogram is shown in Supplementary Figure 2. The 99th percentile was used as the cut point between “high” and “low” volume centers, which relates to approximately 55 surgical cases per year.

Round 2

Reviewer 4 Report

Thank you for the revisions. They strengthen the manuscript and the paper is now acceptable for publication. One small comment: for Supplementary Table 1, we can see that the vast majority of patients >70 did not receive chemotherapy. Therefore because of the small sample size I would add a disclaimer about this that even though the most benefit was in the age >70 group a very small percentage of patients received chemo

Author Response

We thank Reviewer #4 again for the review of our manuscript. The reviewer is correct that the majority of patients >70 years in age do not receive chemotherapy, and we agree that the results of this older cohort receiving chemotherapy should be interpreted with caution. 

The manuscript includes a disclaimer in the discussion on this point (lines 202-205):

"One finding on our subgroup analysis that may be worth further exploration is the association with increased OS among elderly and African American patients receiving multi-agent chemotherapy. Caution is needed in interpreting this result, as the subgroup sizes are small and may be especially prone to selection biases."

We have also highlighted the small sample size of this cohort in the results section (lines 149-153):

"Of note, younger patients were not more likely to benefit from multi-agent chemotherapy; in fact, we observed that for patients ≥70 years of age and for African American patients, multi-agent chemotherapy was associated with improved survival. However, based on small sample size, these factors were not considered in defining a subgroup of patients most likely to benefit from chemotherapy and the subsequent Kaplan-Meier analysis in Figure 4."